# Meta-Learning Requires Meta-Augmentation

**Janarthanan Rajendran**[*][†]
University of Michigan
rjana@umich.edu

**Alex Irpan**[*]
Google Brain
alexirpan@google.com

**Eric Jang**[*]
Google Brain
ejang@google.com

## Abstract

Meta-learning algorithms aim to learn two components: a model that predicts targets for a task, and a base learner that updates that model when given examples from a new task. This additional level of learning can be powerful, but it also creates another potential source of overfitting, since we can now overfit in either the model or the base learner. We describe both of these forms of meta-learning overfitting, and demonstrate that they appear experimentally in common meta-learning benchmarks. We introduce an information-theoretic framework of meta-augmentation, whereby adding randomness discourages the base learner and model from learning trivial solutions that do not generalize to new tasks. We demonstrate that meta-augmentation produces large complementary benefits to recently proposed meta-regularization techniques.

## 1 Introduction

In several areas of machine learning, data augmentation is critical to achieving state-of-the-art performance. In computer vision [31], speech recognition [17], and natural language processing [8], augmentation strategies effectively increase the support of the training distribution, improving generalization. Although data augmentation is often easy to implement, it has a large effect on performance. For a ResNet-50 model [13] applied to the ILSVRC2012 object classification benchmark [28], removing randomly cropped images and color distorted images from the training data results in a 7% reduction in accuracy (see Appendix B). Recent work in reinforcement learning [18, 21] also demonstrate large increases in reward from applying image augmentations.

Meta-learning has emerged in recent years as a popular framework for learning new tasks in a sample-efficient way. The goal is to learn how to learn new tasks from a small number of examples, by leveraging knowledge from learning other tasks [5, 7, 14, 30, 33]. Given that meta-learning adds an additional level of model complexity to the learning problem, it is natural to suspect that data augmentation plays an equally important - if not greater - role in helping meta-learners generalize to new tasks. In classical machine learning, data augmentation turns one example into several examples. In meta-learning, meta-augmentation turns one task into several tasks. Our key observation is that given labeled examples $(x, y)$, classical data augmentation adds noise to inputs $x$ without changing $y$, but for meta-learning we must do the opposite: add noise to labels $y$, without changing inputs $x$.

The main contributions of our paper are as follows: first, we present a unified framework for meta-data augmentation and an information theoretic view on how it prevents overfitting. Under this framework, we interpret existing augmentation strategies and propose modifications to handle two forms of overfitting: *memorization overfitting*, in which the model is able to overfit to the training set without relying on the learner, and *learner overfitting*, in which the learner overfits to the training set and does not generalize to the test set. Finally, we show the importance of meta-augmentation on a variety of benchmarks and meta-learning algorithms.

---

[*]Equal contribution.
[†]This work was done when the author was an intern at Google Brain.

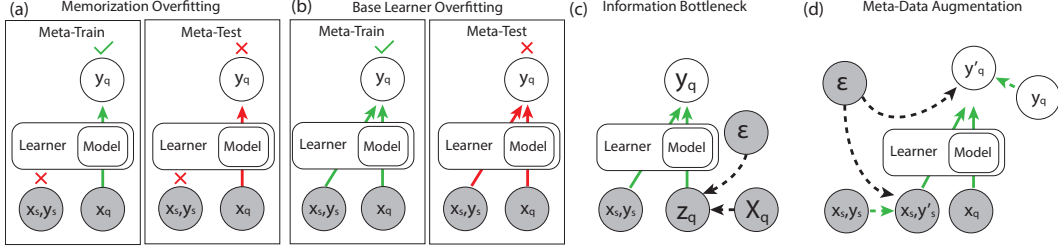

Figure 1: Meta-learning problems provide support inputs $(x_s, y_s)$ to a *base learner*, which applies an update to a *model*. Once applied, the model is given query input $x_q$, and must learn to predict query target $y_q$. **(a)** Memorization overfitting occurs when the base learner and $(x_s, y_s)$ does not impact the model's prediction of $y_q$. **(b)** Learner overfitting occurs when the model and base learner leverage both $(x_s, y_s)$ and $x_q$ to predict $y_q$, but fails to generalize to the meta-test set. **(c)** Yin et al. [37] propose an information bottleneck constraint on the model capacity to reduce memorization overfitting. **(d)** To tackle both forms of overfitting, we view meta-data augmentation as widening the task distribution, by encoding additional random bits $\epsilon$ in $(x_s, y_s)$ that must be decoded by the base learner and model in order to predict a transformed $y'_q$.

## 2   Background and definitions

A standard supervised machine learning problem considers a set of training data $(x^i, y^i)$ indexed by $i$ and sampled from a task $\mathcal{T}$, where the goal is to learn a function $x \mapsto \hat{y}$. In meta-learning, we have a set of tasks $\{\mathcal{T}^i\}$, where each task $\mathcal{T}^i$ is made of a *support* set $(x_s, y_s)$, and a *query* set $(x_q, y_q)$. The grouped support and query sets are referred to as an *episode*. The training and test sets of examples are replaced by *meta-training* and *meta-test* sets of tasks, each of which consist of episodes. The goal is to learn a *base learner* that first observes support data $(x_s, y_s)$ for a new task, then outputs a *model* which yields correct predictions $\hat{y}_q$ for $x_q$. When applied to classification, this is commonly described as $k$-shot, $N$-way classification, indicating $k$ examples in the support set, with class labels $y_s, y_q \in 1, \ldots, N$. Following Triantafillou et al. [34], we rely mostly on meta-learning centric nomenclature but borrow the terms "support", "query", and "episode" from the few-shot learning literature.

Meta-learning has two levels of optimization: an *inner-loop* optimization and an *outer-loop* optimization. The *inner-loop* optimization (the base learner) updates a model using $(x_s, y_s)$. In the *outer-loop*, the updated model is used to predict $\hat{y}_q$ from $x_q$. The *outer-loop* optimization updates the base learner in order to improve how the inner-loop update is carried out. While many meta-learning benchmarks design the support set $(x_s, y_s)$ to have the same data type as the query set $(x_q, y_q)$, the support need not match the data type of the query and can be any arbitrary data. Some meta-learners utilize an inner-loop objective that bears no resemblance to the objective computed on the query set [10, 24]. Contextual meta-learners replace explicit inner-loop optimization with information extraction using a neural network, which is then learnt jointly in the outer-loop optimization [11, 29]. As shown in Figure 1, meta-learners can be treated as black-box functions mapping two inputs to one output, $f : ((x_s, y_s), x_q) \to \hat{y}_q$. We denote our support and query sets as $(x_s, y_s)$ and $(x_q, y_q)$, as all of our tasks have matching inner and outer-loop data types.

Overfitting in the classical machine learning sense occurs when the model has memorized the specific instances of the training set at the expense of generalizing to new instances of the test set. In meta-learning, there are two types of overfitting that can happen: 1) memorization overfitting and 2) learner overfitting. We first discuss memorization overfitting.

A set of tasks are said to be *mutually-exclusive* if a single model cannot solve them all at once. For example, if task $\mathcal{T}^1$ is "output 0 if the image is a dog, and 1 if it is a cat", and task $\mathcal{T}^2$ is "output 0 if the image is a cat, and 1 if it is a dog", then $\{\mathcal{T}^1, \mathcal{T}^2\}$ are mutually-exclusive. A set of tasks is *non-mutually-exclusive* if the opposite is true: one model can solve all tasks at once. Yin et al. [37] identify the memorization problem in meta-learning that can happen when the set of tasks are non-mutually-exclusive. A non-mutually-exclusive setting potentially leads to *complete meta-learning memorization*, where the model memorizes the support set and predicts $\hat{y}_q$ using only $x_q$, without relying on the base learner (shown in Figure 1a). This is acceptable for training

performance, but results in poor generalization on the meta-test set, because the memorized model does not have knowledge of test-time tasks and does not know how to use the base learner to learn the new task. We refer to this as "memorization overfitting" for the remainder of the paper. For example, in MAML [9], memorization overfitting happens when the learned initialization memorizes all the training tasks and does not adapt its initialization for each task using $(x_s, y_s)$. In CNP [11], it occurs when the prediction does not depend on information extracted in the learned latent $z$ from $(x_s, y_s)$. An mutually-exclusive set of tasks is less prone to memorization overfitting since the model must use $(x_s, y_s)$ to learn how to adapt to each task during training.

In a $k$-shot, $N$-way classification problem, we repeatedly create random tasks by sampling $N$ classes from the pool of all classes. Viewing this as a meta-augmentation, we note this creates a mutually-exclusive set of tasks, since class labels will conflict across tasks. For example, if we sampled $\mathcal{T}_1$ and $\mathcal{T}_2$ from the earlier example, the label for dog images is either $0$ or $1$, depending on the task. Yin et al. [37] argue it is not straightforward to replicate this recipe in tasks like regression. Yin et al. [37] propose applying weight decay and an information bottleneck to the model (Figure 1c), which restricts the information flow from $x_q$ and the model to $y_q$. They construct meta-regularized (MR) variants of contextual (CNP [11]) and gradient-based (MAML [9]) meta-learners and demonstrate its effectiveness on a novel non-mutually-exclusive pose regression dataset. In the proceeding sections, we will show that an alternative randomized augmentation can be applied to this dataset, and other such non-mutually-exclusive meta-learning problems.

Learner overfitting (shown in Figure 1b) happens when the base learner overfits to the training set tasks and does not generalize to the test set tasks. The learned base learner is able to leverage $(x_s, y_s)$ successfully to help the model predict $y_q$ for each episode in the meta-training set, coming from the different training set tasks. But, the base learner is unable to do the same for novel episodes from the meta-test set. Learner overfitting can happen both in non-mutually-exclusive and mutually-exclusive task settings, and can be thought of as the meta-learning version of classical machine learning overfitting.

# 3 Meta-augmentation

Let $X, Y$ be random variables representing data from a task, from which we sample examples $(x, y)$. An **augmentation** is a process where a source of independent random bits $\epsilon$, and a mapping $f : \epsilon, X, Y \to X, Y$, are combined to create new data $X', Y' = f(\epsilon, X, Y)$. We assume all $(x, y) \in (X, Y)$ are also in $(X', Y')$. An example is where $\epsilon \sim U[0, 2\pi]$, and $f$ rotates image $x$ by $\epsilon$, giving $X' = rotations(X), Y' = Y$. We define an augmentation to be **CE-preserving** (conditional entropy preserving) if conditional entropy $\mathcal{H}(Y'|X') = \mathcal{H}(Y|X)$ is conserved; for instance, the rotation augmentation is CE-preserving because rotations in $X'$ do not affect the predictiveness of the original or rotated image to the class label. CE-preserving augmentations are commonly used in image-based problems [22, 31, 34]. Conversely, an augmentation is **CE-increasing** if it increases conditional entropy, $\mathcal{H}(Y'|X') > \mathcal{H}(Y|X)$. For example, if $Y$ is continuous and $\epsilon \sim U[-1, 1]$, then $f(\epsilon, x, y) = (x, y + \epsilon)$ is CE-increasing, since $(X', Y')$ will have two examples $(x, y_1), (x, y_2)$ with shared $x$ and different $y$, increasing $\mathcal{H}(Y'|X')$. These are shown in Figure 2.

The information capacity of a learned channel is given by $I(X; \hat{Y}|\theta, \mathcal{D})$, the conditional mutual information given model parameters $\theta$ and meta-training dataset $\mathcal{D}$. Yin et al. [37] propose using a Variational Information Bottleneck [2] to regularize the model by restricting the information flow between $x_q$ and $y_q$. Correctly balancing regularization to prevent overfitting or underfitting can be challenging, as the relationship between constraints on model capacity and generalization are hard to predict. For example, overparameterized networks have been empirically shown to have lower generalization error [25]. To corroborate the difficulty of understanding this relationship, in Section 5.2 we show that weight decay limited the baseline performance of MAML on the Pascal3D pose regression task, and when disabled it performs substantially better.

Rather than crippling the model by limiting its access to $x_q$, we want to instead use data augmentation to encourage the model to pay more attention to $(x_s, y_s)$. A naive approach would augment in the same way as classical machine learning methods, by applying a CE-preserving augmentation to each task. However, the overfitting problem in meta-learning requires different augmentation. We wish to couple $(x_s, y_s), (x_q, y_q)$ together such that the model cannot minimize training loss using $x_q$ alone. This can be done through CE-increasing augmentation. Labels $y_s, y_q$ are encrypted to $y_s', y_q'$ with the

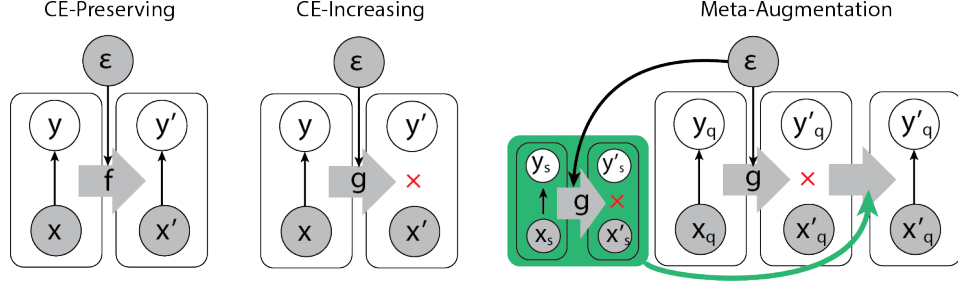

Figure 2: **Meta-augmentation:** We introduce the notion of *CE-preserving* and *CE-increasing* augmentations to explain why meta augmentation differs from standard data augmentation. Given random variables $X, Y$ and an external source of random bits $\epsilon$, we augment with a mapping $f(\epsilon, X, Y) = (X', Y')$. **Left:** An augmentation is *CE-preserving* if it preserves conditional entropy between $x, y$. **Center:** A *CE-increasing* augmentation increases $\mathcal{H}(Y'|X')$. **Right:** Invertible CE-increasing augmentations can be used to combat memorization overfitting: the model must rely on the base learner to implicitly recover $\epsilon$ from $x'_s, y'_s$ in order to restore predictiveness between the input and label.

same random key $\epsilon$, in a way such that the base learner can only recover $\epsilon$ by associating $x_s \rightarrow y'_s$, and doing so is necessary to associate $x_q \rightarrow y'_q$. See Figure 1d and Figure 2 for a diagram.

Although CE-increasing augmentations may use any $f$, within our experiments we only consider CE-increasing augmentations where $f(\epsilon, x, y) = (x, g(\epsilon, y))$ and $g : \epsilon, Y \rightarrow Y$ is invertible given $y$. We do this because our aim is to increase task diversity. Given a single task $\mathcal{T}$, such CE-increasing augmentations create a distribution of tasks $\{\mathcal{T}^\epsilon\}$ indexed by $\epsilon$, and assuming a fixed noise source $\epsilon$, said distribution is widest when $g$ is invertible. CE-increasing augmentations of this form raise $\mathcal{H}(Y'_q|X_q)$ by $\mathcal{H}(\epsilon)$. We state this more formally below.

**Theorem 1.** *Let $\epsilon$ be a noise variable independent from $X, Y$, and $g : \epsilon, Y \rightarrow Y$ be the augmentation function. Let $y' = g(\epsilon, y)$, and assume that $(\epsilon, x, y) \mapsto (x, y')$ is a one-to-one function. Then $\mathcal{H}(Y'|X) = \mathcal{H}(Y|X) + \mathcal{H}(\epsilon)$.*

A proof is in Appendix A. In order to lower $\mathcal{H}(Y'_q|X_q)$ to a level where the task can be solved, the learner must extract at least $\mathcal{H}(\epsilon)$ bits from $(x_s, y'_s)$. This reduces memorization overfitting, since it guarantees $(x_s, y'_s)$ has some information required to predict $y'_q$, even given the model and $x_q$.

CE-increasing augmentations move the task setting from non-mutually-exclusive to mutually-exclusive, as once the data is augmented, no single model can solve all tasks at once. In MAML, with CE-increasing augmentation, the learned initialization can no longer memorize all the training tasks, as for a given input $x$ there could be different correct outputs $y$ in the training data. In CNPs, the learned latent $z$ cannot be ignored by the final conditional decoder, again because each $x$ can have several correct outputs $y$ depending on the task. In both cases, the model is forced to depend on the support set $(x_s, y'_s)$ of each task every time during training to figure out the correct $y'_q$ for $x_q$, thereby making the method less prone to memorization overfitting.

By adding new and different varieties of tasks to the meta-train set, CE-increasing augmentations also help avoid learner overfitting and help the base learner generalize to test set tasks. This effect is similar to the effect that data augmentation has in classical machine learning to avoid overfitting.

Few shot classification benchmarks such as Mini-ImageNet [36] have meta-augmentation in them by default. New tasks created by shuffling the class index $y$ of previous tasks are added to the training set. Here, $y' = g(\epsilon, y)$, where $\epsilon$ is a permutation sampled uniformly from all permutations $S_N$. The $\epsilon$ can be viewed as an encryption key, which $g$ applies to $y$ to get $y'$. This augmentation is CE-increasing, since given any initial label distribution $Y$, augmenting with $Y' = g(\epsilon, Y)$ gives a uniform $Y'|X$. The uniform distribution has maximal conditional entropy, so CE must increase unless $Y|X$ was already uniform. The information required to describe $Y'|X$ has now increased, due to the unknown $\epsilon$. Therefore, this makes the task setting mutually-exclusive, thereby reducing memorization overfitting. This is accompanied by creation of new tasks through combining classes from different tasks, adding more variation to the meta-train set. These added tasks help avoid learner overfitting, and in Section 5.1 we analyze the size of this effect.

Our formulation generalizes to other forms of CE-increasing augmentation besides permuting class labels for classification. For multivariate regression tasks where the support set contains a regression target, the dimensions of $y_s, y_q$ can be treated as class logits to be permuted. This reduces to an identical setup to the classification case. For scalar meta-learning regression tasks and situations where output dimensions cannot be permuted, we show the CE-increasing augmentation of adding uniform noise to the regression targets $y_s' = y_s + \epsilon, y_q' = y_q + \epsilon$ generates enough new tasks to help reduce overfitting.

## 4 Related work

Data augmentation has been applied to several domains with strong results, including image classification [19], speech recognition [17], reinforcement learning [18], and language learning [39]. Within meta-learning, augmentation has been applied in several ways. Mehrotra and Dukkipati [23] train a generator to generate new examples for one-shot learning problems. Santoro et al. [29] augmented Omniglot using random translations and rotations to generate more examples within a task. Liu et al. [22] applied similar transforms, but treated it as task augmentation by defining each rotation as a new task. These augmentations add more data and tasks, but do not turn non-mutually-exclusive problems into mutually-exclusive ones, since the pairing between $x_s, y_s$ is still consistent across meta-learning episodes, leaving open the possibility of memorization overfitting. Antoniou and Storkey [3] and Khodadadeh et al. [16] generate tasks by randomly selecting $x_s$ from an unsupervised dataset, using data augmentation on $x_s$ to generate more examples for the random task. We instead create a mutually-exclusive task setting by modifying $y_s$ to create more tasks with shared $x_s$.

The large interest in the field has spurred the creation of meta-learning benchmarks [20, 27, 34, 36, 38], investigations into tuning few-shot models [4], and analysis of what these models learn [26]. For overfitting in MAML in particular, regularization has been done by encouraging the model's output to be uniform before the base learner updates the model [15], limiting the updateable parameters [40], or regularizing the gradient update based on cosine similarity or dropout [12, 35]. Our work is most closely related to Yin et al. [37], which identifies the non-mutually-exclusive tasks problem, and proposes an information bottleneck to address memorization overfitting. Our paper tackles this problem through appropriately designed meta-augmentation.

## 5 Experiments

We ablate meta-augmentation across few-shot image classification benchmarks in Section 5.1, demonstrating the drop in performance when these benchmarks are made non-mutually-exclusive by removing the meta-augmentation present in them. Section 5.2 then demonstrates gains achievable from using CE-increasing augmentations on regression datasets.[3]

### 5.1 Few-shot image classification (Omniglot, Mini-ImageNet, D'Claw)

We carry out our experiments on $k$-shot $N$-way classification tasks where $k = 1, N = 5$. Common few-shot image classification benchmarks, like Omniglot and Mini-ImageNet, are already mutually-exclusive by default through meta-augmentation. In order to study the effect of meta-augmentation using task shuffling on various datasets, we turn these mutually-exclusive benchmarks into non-mutually-exclusive versions of themselves by partitioning the classes into groups of $N$ classes without overlap. These groups form the meta-train tasks, and over all of training, class order is never changed.

**Omniglot** The Omniglot dataset [20] is a collection of 1623 different handwritten characters from different alphabets. The task is to identify new characters given a few examples of that character. We train a 1-shot, 5-way classification model using MAML [9]. When early stopping is applied, all our MAML models reached 98% test-set accuracy, including on non-mutually-exclusive task sets. This suggests that although non-mutually-exclusive meta-learning problems allow for memorization overfitting, it is still possible to learn models which do not overfit and generalizes to novel test-time tasks if there is sufficient training data. We therefore moved to more challenging datasets.

**Mini-ImageNet** Mini-ImageNet [36] is a more complex few-shot dataset, based on the ILSVRC object classification dataset [6]. There are 100 classes in total, with 600 samples each. We train 1-shot, 5-way classification models using MAML. We ran experiments in three variations, shown in Figure 3. In *non-mutually-exclusive*, classes are partitioned into ordered tasks that keep the class index consistent across epochs. In *intershuffle*, tasks are formed by sampling classes randomly, where previously used classes can be resampled in later tasks. This is the default protocol for Mini-ImageNet. Doing so ensures that over the course of training, a given image $x_q$ will be assigned multiple different class indices $y_q$, making the task setting mutually-exclusive. This also creates more diverse groups of classes / tasks compared to the previous partitioning method. To ablate whether diversity or mutual exclusivity is more important, we devise the *intrashuffle* scheme, which shuffles class label order within the partitions from the non-mutually-exclusive setup. This uses identical classes within each task to non-mutually-exclusive, and the only difference is varying class order within those tasks. This again gives a mutually-exclusive setting, but with less variety compared to intershuffle.

Table 1 (Mini-ImageNet (MAML)) shows that non-mutually-exclusive does worst, followed by intrashuffle, then intershuffle. There is an increase in accuracy from 30% to 43% between non-mutually-exclusive and intrashuffle, and only 43% to 46% between intrashuffle and intershuffle, which indicates shuffled task order is the more important factor. The gap between intrashuffle and intershuffle also indicates that fully randomizing the batches of tasks performs slightly better, as each class is compared to more diverse sets of negative classes. We emphasize that if shuffling is applicable to the problem, intershuffle is the recommended approach. Interestingly, Figure 4 shows that although all non-mutually-exclusive runs overfit at training time, whether they exhibit memorization overfitting or learner overfitting depends on random seed. Both forms of overfitting are avoided by meta-augmentation through intershuffle.

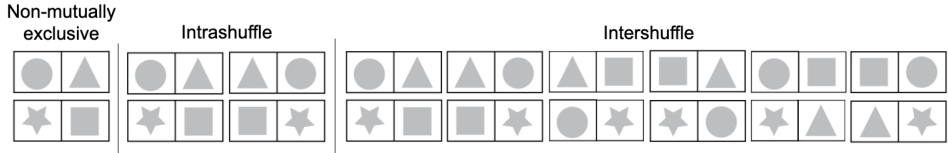

Figure 3: **Non-mutually-exclusive, intrashuffle, and intershuffle.** In this example, the dataset has 4 classes, and the model is a 2-way classifier. In non-mutually-exclusive, the model always sees one of 2 tasks. In intrashuffle, the model sees permutations of the classes in the non-mutually-exclusive tasks, which changes class order. In intershuffle, the model sees $4 \times 3 = 12$ tasks.

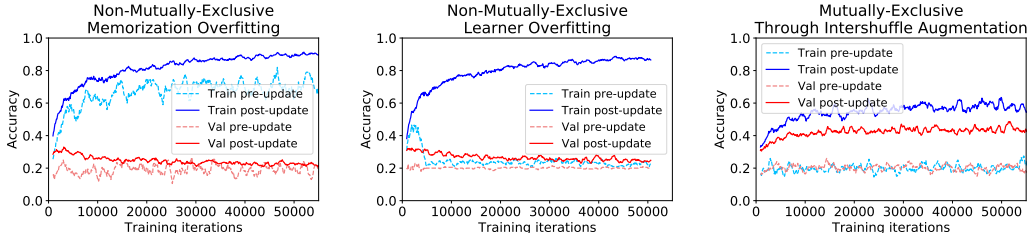

Figure 4: Mini-ImageNet results with MAML. **Left:** In a non-mutually-exclusive setting, this model exhibits memorization overfitting. Train-time performance is high, even before the base learner updates the model based on $(x_s, y_s)$, indicating the model pays little attention to $(x_s, y_s)$. The model fails to generalize to the held-out validation set. **Center:** This model exhibits learner overfitting. The gap between train pre-update and train post-update indicates the model does pay attention to $(x_s, y_s)$, but the entire system overfits and does poorly on the validation set. The only difference between the left and center plots is the random seed. **Right:** With intershuffle augmentation, the gap between train pre-update and train post-update indicates the model pays attention to $(x_s, y_s)$, and higher train time performance lines up with better validation set performance, indicating less overfitting.

**D'Claw dataset** So far, we have only examined CE-increasing augmentations in existing datasets. To verify the phenomenon applies to a newly generated dataset, we collected a small robotics-inspired image classification dataset using hardware designs from ROBEL [1]. Figure 5 shows a D'Claw mounted inside a D'Lantern. An object is placed between the fingers of the claw, and the task is to

Table 1: Few-shot image classification test set results. Results use MAML unless otherwise stated. All results are in 1-shot 5-way classification, except for D'Claw which is 1-shot 2-way.

| Problem setting | Non-mutually-exclusive accuracy | Intrashuffle accuracy | Intershuffle accuracy |
|---|---|---|---|
| Omniglot | 98.1% | 98.5% | **98.7%** |
| Mini-ImageNet (MAML) | 30.2% | 42.7% | **46.0%** |
| Mini-ImageNet (Prototypical) | 32.5% | 32.5% | **37.2%** |
| Mini-ImageNet (Matching) | 33.8% | 33.8% | **39.8%** |
| D'Claw | 72.5% | 79.8% | **83.1%** |

classify whether that object is in the proper orientation or not. A classifier learned on this dataset could be used to automatically annotate whether an autonomous robot has correctly manipulated an object. Labels are $y \in \{0, 1\}$, where 0 means wrong orientation and 1 means correct orientation. The dataset has 20 different classes of 35 images each, and the problem is set up as a 1-shot, 2-way classification problem. We see a change from $72.5\%$ to $83.1\%$ for MAML between non-mutually-exclusive and intershuffle, a gain consistent with the Mini-ImageNet results.

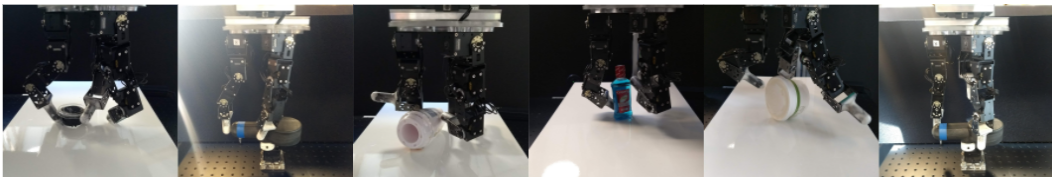

Figure 5: Example images from the D'Claw classification task. An object is placed between the fingers of the claw. Each object has a target orientation, and the model must classify whether the object is in the correct orientation or not.

**Other Meta-Learners** The problem of non-mutually-exclusive tasks is not just a problem for MAML – it is a general problem for any algorithm that jointly trains a learning mechanism and model. Applying the same non-mutually-exclusive Mini-ImageNet transform, we evaluate baseline algorithms from Meta-Dataset [34], and show similar drops in performance for their implementation of Matching Networks [36] and Prototypical Networks [32] (Table 1). For these methods, performance is identical between non-mutually-exclusive and intrashuffle because they are nearest-neighbor based models, which are permutation invariant by construction. We also consider CNP [11] in Section (5.2).

## 5.2 Regression tasks (Sinusoid, Pascal3D Pose Regression)

Our regression experiments use tasks with a scalar output, which are not well suited to shuffling. For these tasks, we experiment with the CE-increasing augmentation of simply adding randomly sampled noise to regression targets $y'_s = y_s + \epsilon, y'_q = y_q + \epsilon$.

**Sinusoid** We start with the toy 1D sine wave regression problem, where $y = A\sin(x - \phi)$. Amplitude $A$ and phase $\phi$ are sampled as $A \sim U[0.1, 5.0]$ and $\phi \sim U[0, \pi]$. To create a non-mutually-exclusive version, domain $[-5, 5]$ is subdivided into 10 disjoint intervals $[-5, -4.5], [-4, -3.5], \ldots, [4, 4.5]$. Each interval is assigned a different task $(A, \phi)$. The inputs $x$ are sampled from these disjoint intervals and $(x, y)$ is generated based on the sine wave for the interval $x$ lies in. The 10 tasks are non-mutually-exclusive because the gaps between intervals means there is a continuous function over $[-5, 5]$ that exactly matches the piecewise function over the sub-intervals where the piecewise function is defined. The function is not unique outside the sub-intervals, but at evaluation time $x$ is always sampled within the sub-intervals.

To augment MAML, for each task $A, \phi$ and the corresponding $x$ interval, new tasks $(x, y + \epsilon)$ where $\epsilon \sim U[-1, 1]$ are added to the meta-train dataset. This is CE-increasing, since even given $x$ and thereby the interval it lies in, the model can no longer uniquely identify the task, thereby increasing $\mathcal{H}(Y|X)$. Figure 6a compares the performance of models that are 1) trained with supervised regression on all the tasks 2) trained using MAML and 3) trained using MAML + augmentation.

We observe that MAML + augmentation performs best. Since the tasks are non-mutually-exclusive, the baseline MAML model memorizes all the different training tasks (sine functions). The 1D sine regression problem is simple enough that MAML can learn even test tasks to some extent with a few gradient steps, but MAML + augmentation adapts better and more quickly because of less memorization.

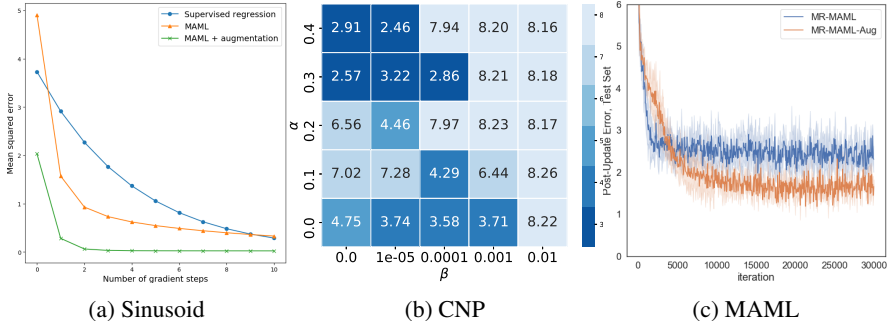

| (a) Sinusoid | (b) CNP | (c) MAML |

Figure 6: **(a)** We add CE-increasing noise to $y_q$ such that the noise must be inferred from the support set, and show that it improves generalization on a sinusoid regression task. **(b)** Post-update MSE on the meta-test set for MR-CNP, sweeping combinations of model regularization strength $\beta$ and meta-data augmentation strength $\alpha$. Lower MSE is better. **(c)** Meta-Augmentation with noise sampled from a discrete set $\alpha \in \{0, .25, .5, .75\}$ provides additional improvements on top of MR-MAML.

**Pascal3D Pose Regression** We show that for the regression problem introduced by Yin et al. [37], it is still possible to reduce overfitting via meta augmentation. We extended the open-source implementation provided by Yin et al. [37] of their proposed methods, MR-MAML and MR-CNP and show that meta augmentation provides large improvements on top of IB regularization. Each task is to take a 128x128 grayscale image of an object from the Pascal 3D dataset and predict its angular orientation $y_q$ (normalized between 0 and 10) about the Z-axis, with respect to some unobserved canonical pose specific to each object. The set of tasks is non-mutually-exclusive because each object is visually distinct, allowing the model to overfit to the poses of training objects, neglecting the base learner and $(x_s, y_s)$. Unregularized models have poor task performance at test time, because the model does not know the canonical poses of novel objects.

We use two different augmentation schemes: for CNP models, we added random uniform noise in the range $(-10\alpha, 10\alpha)$ to both $y_s$ and $y_q$, with angular wraparound to the range $(0, 10)$. To study the interaction effects between meta augmentation and the information bottleneck regularizer, we did a grid search over noise scale $\alpha \in \{0, 0.1, \ldots, 0.5\}$ from our method and KL penalty weight $\beta \in \{0, 10^{-5}, 10^{-4}, \ldots, 10^{-2}\}$ from [37]. Figure 6b shows that data augmentation's benefits are complementary to model regularization. Adding uniform noise to MAML and MR-MAML resulted in underfitting the data (Appendix C), so we chose a simpler CE-increasing augmentation where $\epsilon$ is randomly sampled from a discrete set $\{0., 0.25, 0.5, 0.75\}$. This fixed the problem of underfitting while still providing regularization benefits. Best results are obtained by applying augmentation to MR-MAML without the IB bottleneck ($\beta = 0$, weights are still sampled stochastically).

Table 2 reports meta-test MSE scores from Yin et al. [37] along with our results, aggregated over 5 independent training runs. The baseline was underfitting due to excessive regularization; the existing MAML implementation is improved by removing the weight decay (WD) penalty. See Appendix C for further discussion.

# 6 Discussion and future work

Setting aside how "task" and "example" are defined for a moment, the most general definition of a meta-learner is a black box function $f : ((x_s, y_s), x_q) \mapsto \hat{y}_q$. In this light, memorization overfitting is just a classical machine learning problem in disguise: a function approximator pays too much attention to one input $x_q$, and not enough to the other input $(x_s, y_s)$, when the former is sufficient to solve the task at training time. The two inputs could take many forms, such as different subsets of pixels within the same image.

Table 2: Pascal3D pose prediction error (MSE) means and standard deviations. Removing weight decay (WD) improves the MAML baseline and augmentation improves the MAML, MR-MAML, CNP, MR-CNP results. Bracketed numbers copied from Yin et al. [37].

| Method | MAML (WD=1e-3) | MAML (WD=0) | MR-MAML ($\beta$=0.001) | MR-MAML ($\beta$=0) | CNP | MR-CNP |
|--------|----------------|-------------|--------------------------|----------------------|-----|--------|
| No Aug | [5.39$\pm$1.31] | 3.74 $\pm$ .64 | 2.41 $\pm$ .04 | 2.8 $\pm$ .73 | [8.48$\pm$.12] | [2.89$\pm$.18] |
| Aug | **4.99$\pm$1.22** | **2.34 $\pm$ .66** | **1.71 $\pm$ .16** | **1.61 $\pm$ .06** | **2.51$\pm$.17** | **2.51$\pm$.20** |

By a similar analogy, learner overfitting corresponds to correct function approximation on input examples $((x_s, y_s), x_q)$ from the training set, and a systematic failure to generalize from those to the test set. Although we have demonstrated that meta-augmentation is helpful, it is important to emphasize it has its limitations. Distribution mismatch between train-time tasks and test-time tasks can be lessened through augmentation, but augmentation may not entirely remove it.

One crucial distinction between data-augmentation in classical machine learning and meta-augmentation in meta-learning is the importance of CE-increasing augmentations. For data-augmentation in classical machine learning, the aim is to generate more varied *examples*, within a single task. Meta-augmentation has the exact opposite aim: we wish to generate more varied *tasks*, for a single *example*, to force the learner to quickly learn a new task from feedback. Our experiments exclusively focused on changing outputs $y$ without changing inputs $x$, as these augmentations were simpler and easier to study. However, given the usefulness of CE-preserving augmentations in classical setups, it is likely that a combination of CE-preserving noise on $x$ and CE-increasing noise on $y$ will do best. We leave this for future work.

## Broader Impact

Our work discusses methods of improving meta-learning through meta-augmentation at the task level, and does not target any specific application of meta-learning. The learning algorithms meta-learning generates are ones learned from the data in train-time tasks. It is possible these approaches inject bias into not only how models perform *after* training, but also inject bias into *the training procedure itself*. Biased learners can result in biased model outcomes even when the support set presented at test time is unbiased, and this may not be straightforward to detect because the behavior of the learner occurs upstream of actual model predictions. We believe our work is a positive step towards mitigating bias in meta-learning algorithms, by helping avoid overfitting to certain parts of the inputs.

## Acknowledgments and Disclosure of Funding

We thank Mingzhang Yin and George Tucker for discussion and help in reproducing experimental results for pose regression experiments. We thank Chelsea Finn and Honglak Lee for early stage project discussion, Erin Grant for help on the meta-dataset codebase, and Luke Metz, Jonathan Tompson, Chelsea Finn, and Vincent Vanhoucke for reviewing publication drafts of the work.

During his internship at Google, Jana devised, implemented and ran MAML experiments for Omniglot, mini-Imagenet and Sinusoid regression. Along with Alex, he collected the ROBEL classification datasets. Alex advised Jana's internship and re-ran Jana's code post-internship to include validation with early stopping. Eric advised Jana's internship and ran experiments on pose prediction and meta-dataset. All three authors interpreted results and contributed to paper writing.

## Footnotes

[3]Code and data available at `https://github.com/google-research/google-research/tree/master/meta_augmentation`.

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
