[Supplementary Material]

# Appendix

## A Proof of $\mathcal{H}(\epsilon)$ increase

**Theorem 1.** *Let $\epsilon$ be a noise variable independent from $X, Y$, and $g : \epsilon, Y \to Y$ be the augmentation function. Let $y' = g(\epsilon, y)$, and assume that $(\epsilon, x, y) \mapsto (x, y')$ is a one-to-one function. Then $\mathcal{H}(Y'|X) = \mathcal{H}(Y|X) + \mathcal{H}(\epsilon)$.*

*Proof.* First, by chain rule of conditional entropy, we have $\mathcal{H}(Y'|X) = \mathcal{H}(X, Y') - \mathcal{H}(X)$ and $\mathcal{H}(Y|X) = \mathcal{H}(X, Y) - \mathcal{H}(X)$. Showing $\mathcal{H}(X, Y') = \mathcal{H}(X, Y) + \mathcal{H}(\epsilon)$, then subtracting $\mathcal{H}(X)$ from both sides is enough to prove the desired statement.

Given any $(x, y')$, by the one-to-one assumption, there is exactly one inverse $(\epsilon, x, y)$. This gives $p(x, y') = p(\epsilon, x, y)$. Since $\epsilon$ is independent of $(x, y)$, we have $p(x, y') = p(\epsilon)p(x, y)$, so $\mathcal{H}(X, Y') = \mathbb{E}\left[-\log p(x, y')\right] = \mathbb{E}\left[-\log p(\epsilon) - \log p(x, y)\right] = \mathcal{H}(\epsilon) + \mathcal{H}(X, Y)$. $\square$

In Theorem 1, we stated that if $(\epsilon, x, y) \mapsto (x, y')$ is one-to-one, then $\mathcal{H}(Y'|X)$ increases by $\mathcal{H}(\epsilon)$. If $\mathcal{H}(Y|X)$ is less than $\mathcal{H}(\epsilon)$ away from the maximum entropy distribution, the one-to-one condition cannot be satisfied, and the theorem does not apply. For example, if $Y$ were $\{0, 1\}$ labels for binary classification, and $\mathcal{H}(Y|X) = 1$, then it would be impossible for $\mathcal{H}(Y'|X)$ to be any larger, because 1 is the maximum entropy distribution over 2 discrete outcomes. This is generally not a concern in practice, since standard ML setups have exactly one label $y$ per example $x$. Such problems have $\mathcal{H}(Y|X) = 0$, leaving room for augmentation to increase $\mathcal{H}(Y'|X)$.

## B ImageNet augmentation ablation

ILSVRC2012 classification experiments were trained on a TPU v2-128, using publicly available source code from `https://github.com/google/flax/tree/master/examples/imagenet`. This codebase applies random crops and random left-right flips to the training images. Training a ResNet-50 baseline achieves 76% top-1 accuracy after 60 epochs. The same model with augmentations removed achieves 69% accuracy. This quantifies the performance increase data augmentation provides.

## C Pose Regression

We modified the open-source implementation provided at `https://github.com/google-research/google-research/tree/master/meta_learning_without_memorization`. Instead of re-generating a validation set from scratch using pose_data code, we used 10% of the training dataset as a validation dataset to determine optimal noise hyperparameters and outer learning rate while keeping the remaining default hyperparameters fixed. Afterward, the validation set was merged back into the training set.

We encountered difficulty reproducing the experimental results for MR-MAML (2.26 (0.09)) from Yin et al. [37], despite using publicly available code and IB hyperparameters suggested by authors via email correspondence. It is possible that the discrepancy was due to another hyperparameter (e.g. outer and inner loop learning rates) not being set properly. We followed the same evaluation protocol as the authors via personal correspondence: for each independent trial, we trained until convergence (the default number of iterations specified) and report the average post-update test error for the last 1000 training iterations. Early stopping on validation error followed by evaluation on the test set may result in better performance across all methods, although we did not do this for the sake of consistency in comparing with prior work.

Adding uniformly sampled noise $U(-10\alpha, 10\alpha)$ hurts MR-MAML performance. We note that this augmentation successfully reduces the gap between training and test error - indicating that it fixes overfitting. However, it seems to result in underfitting on the training set as indicated by lower training set performance. This is shown in Figure 7. We hypothesize that gradients become too noisy under the current architecture, batch size, and learning rate hyperparameters set by the baseline when too much noise is added. Sampling from a discrete set of 4 noise values $\epsilon \in \{0, 0.25, 0.5, 0.75\}$ provides just enough task-level augmentation to combat overfitting without underfitting the training data.

| (a) MR-MAML | (b) Uniform Augmentation | (c) Discrete Augmentation |

Figure 7: **(b)** Adding data augmentation via continuous uniform noise to MR-MAML decreases the train-test gap, but seems to underfit data. We hypothesize that this might be addressed by increasing the size of the model or using an alternate architecture to aid optimization, though we leave model architecture changes out of the scope of this work. Sampling augmentations from a discrete set of noise values **(c)** reduces both training and test error.

To investigate this further, Figure 8a displays test loss as a function of the number of additional noise values $\epsilon$ added to $y_s, y_q$. As the number of discrete noise values becomes large, this becomes approximately equivalent to the uniform noise scheme with $\alpha = 1$. In our experiments, $n$ different noise values is equivalent to increasing $\mathcal{H}(Y'|X)$ by $\log_2 n$, and this can be viewed as quantifying the relationship between added noise and generalization. We find that MR-MAML achieves the lowest error at 4 augmentations, while MAML achieves it at 2 augmentations. As the number of noise values increases, performance gets worse for all methods. This shows it is important to tune the amount of meta-augmentation. MR-MAML exhibits less underfitting than MAML, suggesting that there are complementary benefits to IB regularization and augmentation when more noise is added. Figure 8b shows single-trial test error performance over combinations of how many discrete noise values are added across all IB strength parameters $\beta$. Our best model combines noise with $\beta = 0$, which removes the IB constraint but preserves the stochastic weights used for MAML.

Figure 8: **(a)** Test performance (MSE) as a function of number of discrete noise augmentations. Shaded regions correspond to standard deviation over 5 independent trials. **(b)** Figure 6b, but for MR-MAML (single-trial).

# D  D'Claw data collection

The D'Claw dataset contains images from 10 different objects, placed between the fingers of the claw. Each object had 2 classes, corresponding to whether the object was in the correct natural orientation or not. This task is representative of defining a success detector that might be used as reward in a reinforcement learning context, although we purely treat it as a supervised image classification problem. The task is made more difficult because the claw occludes the view of the object.

The smartphone camera from a Google Pixel 2 XL was used to collect 35 example images for each class. All images were taken from the same point of view in a vertical orientation, but there is small variation in camera location and larger variation in lighting conditions. The images were resized to 84x84 images before training.

## E    Few-shot image classification

All few-shot image classification experiments were run on a cloud machine with 4 NVIDIA Tesla K80s and 32 Intel Broadwell CPUs. To maximize resource usage, four jobs were run at once.

Each experiment was trained for 60000 steps. The validation set accuracy was evaluated every 1000 steps, and the model with best validation set performance was then run on the test set. The code is a modified version of the MAML codebase from `https://github.com/cbfinn/maml`, and uses the same hyperparameters, network architecture, and datasets. These are briefly repeated below.

**Omniglot**    The MAML model was trained with a meta batch size of 32 tasks, using a convolutional model, 1 gradient step in the inner-loop, and an inner learning rate of $\alpha = 0.4$. The problem was set up as a 1-shot 5-way classification problem. The training set contained 1200 characters, the validation set contained 100 characters, and the test set was the remaining 323 characters. Each character had 20 examples.

**Mini-ImageNet**    The MAML model was trained with a meta batch size of 4 tasks, using a convolutional model, 5 gradient steps in the inner loop, and an inner learning rate of $\alpha = 0.01$. The problem was set up as a 1-shot 5-way classification problem. There are 64 training classes, 16 validation classes, and 20 test classes. Each class had 600 examples.

**Meta-Dataset**    Mini-ImageNet experiments from the Meta-Dataset codebase (branched from `https://github.com/google-research/meta-dataset/commit/ece21839d9b3b31dc0e5addd2730605df5dbb991`).   Used the default hyperparameters for Matching Networks and Prototypical Networks.

**D'Claw**    The experiments on D'Claw used the same model architecture as Mini-ImageNet, with a meta batch size of 4 tasks, 5 gradient steps in the inner loop, and an inner learning rate of $\alpha = 0.01$. Unlike the Mini-ImageNet experiments, the problem was set up as a 1-shot 2-way classification problem. For D'Claw experiments, the train set has 6 objects (12 classes), the validation set has 2 objects (4 classes), and the test set has 2 objects (4 classes). Each class had 35 examples. The train, validation, and test splits were generated 3 times. Models were trained on each dataset, and the final reported performance is the average of the 3 test-set performances.