[Reviews · NeurIPS 2020]

Review 1

Summary and Contributions: Summary: This paper studies the problem of memorisation in meta-learning, extending the work of [37] in last ICLR. It discusses two kinds of overfitting that can occur in meta-learning memorisation-overfitting (task model doesn’t use meta-learner’s knowledge) and learner-overfitting (meta-learner’s knowledge doesn’t generalise from meta-train to meta-test). [37] proposed an information bottleneck regularization to reduce overfitting. Here the focus is on meta-augmentation in the form of creating synthetic tasks by adding reversible, conditional entropy-increasing, transformations to the output labels. The first set of experiments investigates classification problems (via omniglot, miniImageNet and D’claw) and explores the memorisation issues discussed in [37] further with a break down between non-mutually exclusive and intershuffle/intrashuffle variants of mutual exclusivity that help to unpack the different kinds of meta-overfitting. The second set of experiments look at a toy regression task and a real 3D pose regression task. The results show that the proposed task augmentation further improve generalisation in a complementary way to [37].

Strengths: Strengths: + Memorization and managing meta-overfitting are important issues to get to grips with for meta-learning research to advance. + The few-shot regression results using label-augmentation provide a solid improvement on prior SOTA [37]. + Enjoyable to read and insightful discussion. + Relevant to the community.

Weaknesses: Weaknesses: 1. The nice introductory discussion in Sec 3 is very nice and insightful. But (i) While it links clearly enough to the regression experiments in Sec 5.2, it could be better liked to the classification experiments in Sec 5.1. How to think about noise providing an encryption key in the latter case? (ii) It would be good to unpack more programatically how a few-shot meta-learner needs to act differently to deal with the noise addition and how this enables it to generalise better. The description “Meta-learner recovers [key/noise] \epsilon by associating x_s -> y’_s in order to associate x_q->y’_q” could be made more concrete in terms of the actual mechanics of the two model types considered (MAML, CNP). 2. The analysis & insight in the classification case doesn’t provide a large extension over [37]. 3. It would be nice to apply the regression part to some more significant problems. EG: few-shot interest point detection problems that are more heavily optimised in vision community, in order to find out if it really makes a benefit to SotA.

Correctness: Probably, yes.

Clarity: OK. But see W1.

Relation to Prior Work: Yes.

Reproducibility: Yes

Additional Feedback: ------ POST FEEDBACK ------- I have read the other reviews and author feedback. Overall the paper is interesting and has insights, so I am above borderline. But I can't give it a higher score on the grounds that the the explanations could be much clearer, especially for such an insight paper, (and regression experiments more substantive).


Review 2

Summary and Contributions: The authors analyze the impact of different meta-augmentation strategies on generalization in meta-learning. They also introduce the concepts of CE-preserving and CE-increasing augmentations, and their effect on task-augmentation in meta-learning.

Strengths: This paper builds on (Yin et al., 2020), and goes in more details as to what make "good" task augmentation strategies. In my opinion, the strongest contribution of the paper is the characterization of CE-preserving and CE-increasing augmentations in the context of meta-learning, and how the former (which is more often used in supervised learning) might not be a prevalent in meta-learning as the latter. I really liked how the authors consider a full spectrum of augmentations (non-mutually exclusive, intrashuffle and intershuffle), which gives a more complete view of the effect of mutually-exclusiveness in meta-learning. I appreciated that the code was provided with the submission. The code is clear, and easy to understand. (Yin et al., 2020) Meta-learning without memorization. Mingzhang Yin, George Tucker, Mingyuan Zhou, Sergey Levine, and Chelsea Finn.

Weaknesses: A. Major concerns 1. I did not understand the experimental setup for the sinusoid experiment in Section 5.2 (lines 238-241). The 10 disjoint intervals do not cover the whole domain [-5, 5]. What happens if x lies in (-4.5, -4) for example? Line 241: "there exists a continuous function that covers each piecewise component."; if I understand correctly the setup, there can't be a continuous function over the whole domain [-5, 5], since you'd end up with discontinuities at the borders of your sub-intervals. An illustration of this task (possibly in the Appendix) would make things clearer. 2. The paper fails to show how much information needs to be retained in the CE-increasing augmentation to still be useful for meta-learning. What happens in the extreme case where the transformation makes X' independent of Y' (e.g. randomly assigning a label Y' to X', instead permuting the labels)? CE-increasing augmentations seem to play an important role with generalization in meta-learning, and this paper clearly shows it, but increasing the conditional entropy does not seem to be a sufficient condition for generalization. I would like to see an experiment that quantifies the effect of the increase in conditional entropy on generalization (not necessarily a full theoretical treatment, but at least an ablation study). 3. Line 89: "overfitting of the inner-loop optimization leads to memorization overfitting". Can you clarify how memorization overfitting corresponds to inner-loop overfitting? Inner-loop overfitting would mean that the base learner would rely *too much* on the support set, as opposed to *too little* in memorization overfitting. I understand that the authors want to emphasize the difference between memorization and learner overfitting, but I believe in both cases this is a pathology at the level of the outer-loop ("inner-loop overfitting" would then correspond to "standard" overfitting on one specific task). B. Moderate concerns 1. Lines 264-265: "regularization strength \beta \in {0, 10−5, 10−4, . . . , 10−2}.". It should be made more explicit that these correspond to the coefficients of the weight decay. 2. Line 131: "CE-increasing augmentations of this form raise H(Y_q | X_q) by H(\epsilon)."; even if this is a simple statement, I would appreciate a small proof (in the Appendix) of this fact for completeness. 3. Lines 68-69: "Such tasks are easier to meta-learn, since the model must use (xs, ys) to learn how to adapt to each task.". It would be more accurate to say "less prone to memorization overfitting" rather than easier to meta-learn. If the authors do indeed mean "easier to meta-learn", then there should be empirical evidence (or a citation) supporting this, since (Yin et al., 2020) does not necessarily show how mutually-exclusive tasks make meta-learning easier (e.g. as in more efficient, faster to converge). 4. Line 142: "[shuffling the class index]. This form of augmentation is CE-increasing". There should be a proof of this statement. C. Minor concerns 1. Line 225: The name of the paragraph "Meta-Dataset" is misleading. I was expecting experiments on Meta-Dataset (like the previous paragraphs read "Omniglot" and "miniImageNet", but these were experiments on some other algorithms (ProtoNet & Matching Networks). These experiments in themselves are great though. 2. Line 179: Typo: "k-shot N-way" 3. In Figure 2, the illustration for CE-increasing augmentation is a bit misleading, because it seems to imply that using a CE-Increasing augmentation leads to Y' being independent of X', which is not necessarily true. 4. In Section 3, this is implicit, but it should be made more explicit that \epsilon is independent of X.

Correctness: D. Correctness The empirical methodology is correct.

Clarity: E. Clarity The paper is clearly written.

Relation to Prior Work: F. Relation to prior work 1. In the "Related work" section, in addition to (Liu et al., 2020, [22] in the submission) which mentions augmentations with rotations, I think it would make sense to also mention how the standard Omniglot dataset was already using these task-augmentations since (Santoro et al., 2016). (Santoro et al., 2016) Adam Santoro, Sergey Bartunov, Matthew Botvinick, Daan Wierstra, Timothy Lillicrap. Meta-Learning with Memory-Augmented Neural Networks. (Liu et al., 2020) Jialin Liu, Fei Chao, and Chih-Min Lin. Task augmentation by rotating for meta-learning

Reproducibility: Yes

Additional Feedback: After authors response -------------------------- I would like to thank the authors for their very clear authors response. After reading it, I feel like all my comments were properly addressed. I am therefore increasing my score to 7.


Review 3

Summary and Contributions: This paper introduces a information theoretic approach to apply data augmentation for few-shot multitask learning.

Strengths: 1. the paper is well written and the motivation is clear 2. the method is tested on several few-shot learning benchmarks

Weaknesses: I'm not an expert on few-shot learning, but the paper does not seem to have many baseline methods that target the overfitting issue.

Correctness: yes they seem correct to me.

Clarity: the paper is well written and clear.

Relation to Prior Work: I'm not very familiar with the related work.

Reproducibility: Yes

Additional Feedback:


Review 4

Summary and Contributions: This paper identifies two forms of overfitting in meta-learning: memorization overfitting and learner overfitting. Memorization overfitting occurs when the meta-learner simply outputs a single model that solves all tasks in the meta-train set while ignoring each support set. Learner overfitting occurs when the meta-learner does incorporate the support set, but in a way that does not generalize to the meta-test set. They proceed to propose an information-theoretic framework for meta-augmentation which tackles both forms of overfitting. --------------------------------Update--------------------------------- I have read the rebuttal along with the reviews. While I think this paper is built upon a novel insight about meta-regularization, I am still unconvinced of the additional value that the experiments in this paper have over [37], or that testing on different augmentation schemes is a sufficient demonstration of the proposed information-theoretic framework. I maintain my score for these reasons, but would also be okay with acceptance.

Strengths: The distinction between CE-preserving and CE-increasing augmentation, along with the claim that we should use CE-increasing augmentation, generalizes the practice of switching labels in few-shot classification tasks. I think this is a novel viewpoint, and that the general formulation of X→X, Y→g(epsilon, Y) will be useful in regularizing supervised meta-learning tasks.

Weaknesses: I found the experimental section underwhelming. All section 5.1 shows is that meta-overfitting occurs when we don't switch labels in few-shot classification, which was one of the main points of [1] and is also commonly known. Section 5.2 proposes the task augmentation y += epsilon using the same random epsilon within each episode. Experiments show that this regularization improves performance. It seems like the only novel augmentation method in this paper is to add a random constant to y in few-shot regression problems, which does not seem like a significant contribution. I feel the paper would have been more convincing if they proposed and evaluated multiple (>= 2) new instantiations of their proposed framework for augmenting tasks y->g(epsilon, y). [1] Mingzhang Yin, George Tucker, Mingyuan Zhou, Sergey Levine, and Chelsea Finn. Meta-learning without memorization.

Correctness: Please refer to comments above

Clarity: Please refer to comments above

Relation to Prior Work: Please refer to comments above

Reproducibility: Yes

Additional Feedback:

[Author Response · NeurIPS 2020]

1 We thank the reviewers for their feedback. We address their comments in the order of R2&R5, R2, R3, and R5.

## REVIEWER #2 AND REVIEWER #5
**#2 3.2, #5 3. Insight in classification in relation to [37]**: Our main contribution of the paper is a framework for thinking about meta-augmentation. Although the importance of label shuffling is known in few-shot classification, we show that these results are consistent with being a special case of our meta-augmentation framework. Also, on top of [37], we go on to show differences between intra-shuffling and inter-shuffling, and that memorization overfitting and learner overfitting are both possible, but not guaranteed to occur for non-mutually-exclusive tasks.

## REVIEWER #2
**3.1.(i). Better linking CE-increasing augmentations to few-shot classification**: In our few shot classification benchmarks, $\epsilon$ is a random variable for a uniformly sampled permutation from $S_N$, and $y' = g(\epsilon, y)$ is the application of the permutation. This augmentation increases the conditional entropy $H(Y_q'|X_q)$ as the amount of information required to describe $Y'$ given $X$ has now increased due to the unknown $\epsilon$. This can be viewed as an encryption key because given just $y'$ we cannot infer $y$, but $y', \epsilon$ recovers $y$ exactly. We will make this link clear and add it in the camera ready version of the paper.

**3.1.(ii). Mechanics of CE-increasing augmentation in MAML and CNP**: We will add an explanation of the actual mechanics in MAML and CNP to the camera ready version of the paper. In general, CE-increasing augmentations force high loss if the meta-learner does not learn to adapt using the support set.

## REVIEWER #3
**3.A.1. Experimental setup of the Sinusoidal regression task**: There was a mistake in our experiment description. In our experiments, $x$ is always sampled from the disjoint intervals $[-5, -4.5], [-4, -3.5], \ldots, [4, 4.5]$, not uniformly from $[-5, 5]$ as mentioned in the paper. So $x$ will never be sampled from $(-4.5, -4)$. The gaps between intervals means there is a continuous function over $[-5, 5]$ that exactly matches the piecewise function over the sub-intervals where the piecewise function is defined. The value of the continuous function outside those intervals can be arbitrary. We will correct this in the camera ready version.

**3.A.2. Quantifying the effect of increase in conditional entropy on generalization** Figure 8a in Appendix B displays test loss as a function of the number of discrete noise values added to $y$. Since we always use augmentations that satisfy the conditions in line 127, this quantifies the $H(Y|X)$ increase as $\log_2 n$ bits, where $n$ is the number of discrete noise values used. All the three methods (CNP, MAML and MR-MAML) follow a $U$-shaped curve with best performance at an intermediate amount of added noise, not too low and not too high. As noted, CE-increasing augmentation is not a sufficient condition for generalization.

**3.A.3. Inner-loop optimization and memorization overfitting**: Yes, we intended memorization overfitting to mean the base learner relying too little on the support set. We will update the wording to clarify this.

**3.B.[1,3]. wording comments:** We agree with the reviewer. We will make $\beta$'s meaning more explicit, and also update lines 68-69 as recommended.

**3.B.2. $H(\epsilon)$ proof**: We will add a proof to the Appendix. We noticed our original statement was not as precise as it should have been. The updated statement follows: Let $\epsilon$ be a noise variable independent from $X, Y$, let $g : \epsilon, Y \to Y$ be the augmentation function. Define $g_\epsilon(y)$ and $g_y(\epsilon)$ as $g(\epsilon, y)$ with $\epsilon$ or $y$ fixed. If $g_\epsilon$ and $g_y$ are one-to-one for all $\epsilon, y$, then $H(Y'|X) = \min(H(Y|X) + H(\epsilon), H(uniform))$. In other words, the CE increases by $H(\epsilon)$, but $H(Y'|X)$ is upper-bounded by the max entropy distribution, the uniform distribution over the codomain. Proof: Ignoring the above edge case, $H(Y'|X) = H(Y, \epsilon|X)$, and independence gives $H(Y, \epsilon|X) = H(Y|X) + H(\epsilon|X) = H(Y|X) + H(\epsilon)$.

**3.B.4 Shuffle CE-increase proof**: We will add this to the appendix. A brief proof sketch follows: let $\epsilon$ be a random variable for a uniformly sampled permutation from $S_N$. Given any initial label distribution, augmenting with $Y' = g(\epsilon, Y)$ gives a uniform $Y'|X$, and since $H(Y'|X)$ is the highest possible conditional entropy, CE must increase unless $Y|X$ was already uniform.

**3.C.[1-4], 6.F.1.**: We agree with all the reviewer's suggestions under "minor concerns", and will also make sure to mention the augmentations used in Omniglot in the camera-ready version.

## REVIEWER #5
**3. Multiple instantiations of proposed framework** We proposed and evaluated two different augmentation methods for pose regression: discrete noise and uniform noise (see Appendix B). We also found 8 different augmentations (shifting, scaling, sign flipping, etc.) to work well on the sinusoid regression task, although we did not include these results for compactness sake, and did not try all 8 on the pose regression task. We believe that the primary contribution of our work lies not in the specific augmentations we used for regression tasks, but in the general information-theoretic framework for meta-augmentation. We demonstrate this framework to be consistent across multiple datasets, models, classification and regression problems, and augmentation strategies.

[Meta-Review · NeurIPS 2020]

This paper addresses the impact of different meta-augmentation strategies, introducing CE-preserving and CE-increasing augmentations and their effect on task-augmentation. During the discussion period, two of reviewers stated that the authors response well addressed their concerns and one reviewer raised his/her score to 7. For the final version, I would to like suggest authors to improve the manuscript such that explanations should be much clearer to make it a solid contribution.